# Vibronic coherence evolution in multidimensional ultrafast photochemical processes

James D. Gaynor [1], Jason Sandwisch [1] & Munira Khalil [1]*

The complex choreography of electronic, vibrational, and vibronic couplings used by photoexcited molecules to transfer energy efficiently is remarkable, but an unambiguous description of the temporally evolving vibronic states governing these processes has proven experimentally elusive. We use multidimensional electronic-vibrational spectroscopy to identify specific time-dependent excited state vibronic couplings involving multiple electronic states, high-frequency vibrations, and low-frequency vibrations which participate in ultrafast intersystem crossing and subsequent relaxation of a photoexcited transition metal complex. We discover an excited state vibronic mechanism driving long-lived charge separation consisting of an initial electronically-localized vibrational wavepacket which triggers delocalization onto two charge transfer states after propagating for ~600 femtoseconds. Electronic delocalization consequently occurs through nonadiabatic internal conversion driven by a 50 cm$^{-1}$ coupling resulting in vibronic coherence transfer lasting for ~1 picosecond. This study showcases the power of multidimensional electronic-vibrational spectroscopy to elucidate complex, non-equilibrium energy and charge transfer mechanisms involving multiple molecular coordinates.

[1] Department of Chemistry, University of Washington, Box 351700 , Seattle, WA 98195, USA. *email: mkhalil@uw.edu

Photoexcited charge and energy transfer are essential steps in many natural and artificial light-harvesting processes[1–4], which often rely on transition metal complexes with characteristically dense manifolds of charge transfer excited states[5–11]. The orchestration of electronic, vibrational, and electronic–vibrational (vibronic) couplings that drive charge and energy transfer in light-harvesting complexes involve structural rearrangement, spin–vibronic interactions, and a dynamic solvent environment making these systems fundamentally interesting and challenging to understand. A wealth of experimental and computational studies on Ru-centered complexes have measured their photoexcited dynamics following singlet metal-to-ligand charge transfer (MLCT) excitation, such as ultrafast intersystem crossing (ISC). These past studies have established that the excited triplet MLCT states can provide sufficiently long-lived charge separation to yield a useful chemical potential for energy harvesting applications[12–18]. Yet, an essential open question remains: what intramolecular coordinates define the trajectory of a photoexcited complex's evolution into these desirable long-lived charge separated states?

Addressing this requires a continually advancing spectroscopic toolbox to move beyond measuring kinetics of photoexcited states toward building detailed molecular-level descriptions of important charge and energy transfer mechanisms utilized by transition metal complexes[19–23]. Exciting progress in experimental techniques is making investigations related to the following questions tractable: how do specific molecular coordinates control photoexcited energy transfer dynamics? What role do vibronic couplings and coherences play during photoexcited energy transfer and relaxation? Third-order nonlinear Fourier transform (FT) techniques such as two-dimensional (2D) electronic (ES) and 2D infrared (IR) spectroscopy have greatly advanced our understanding of coherent molecular phenomena in solution[24–33]. While vibronic information may be measured indirectly by 2D IR and ES spectroscopies, the recently developed 2D electronic–vibrational (2D EV) and 2D vibrational–electronic experiments directly access vibronic information through resonant interactions with electronic and IR-active vibrational transitions[34–37].

In this article, we use multidimensional (2D and 3D) EV spectroscopy to directly follow vibronic coherence evolution in a photoexcited Ru-centered complex: the solar cell dye molecule, [Ru-(dcbpy)$_2$(NCS)$_2$] (dcbpy = 4,4′-dicarboxy-2,2′-bipyridine), commonly known as N3. We demonstrate 3D EV spectroscopy (Fig. 1d), where vibronic couplings between three disparate molecular coordinates are resolved: electronic excitations ($\omega_1$: 24,100–25,500 cm$^{-1}$), low-frequency vibrations ($0 < \omega_2 < 0$–833 cm$^{-1}$), and high-frequency vibrations ($\omega_3$: 1250–1500 cm$^{-1}$). A real-time microscopic view of photochemical reactivity may be obtained by following the temporal evolution of the vibronic states measured in the 3D EV spectrum. As depicted in Fig. 1d, vibrational coherences (red/yellow spheres, L$_1$/L$_1$′) can be directly identified by an electronically localized 3D EV feature, and its time-dependent behavior (L$_1$ (red) → L$_1$′ (yellow)) characterized through its $\omega_2$-dependent spectral phase. Population and coherence transfer (blue and purple spheres, L$_2$) between vibronic states can follow internal conversion (IC) from the perspective of specific vibrational coordinates that may be directly involved. Vibronic coherences can also be identified through out-of-phase oscillatory dynamics between different 3D EV features. In this way, the 3D FT experiment can unveil additional dark coordinates actively coupling the two motions measured in the 2D experiment, as well as coherences between molecular states facilitating energy transfer. While 3D FT studies have yielded exciting insight related to vibrations driving singlet fission[38], quantum coherence in photosynthesis[39], and tracking excited-state

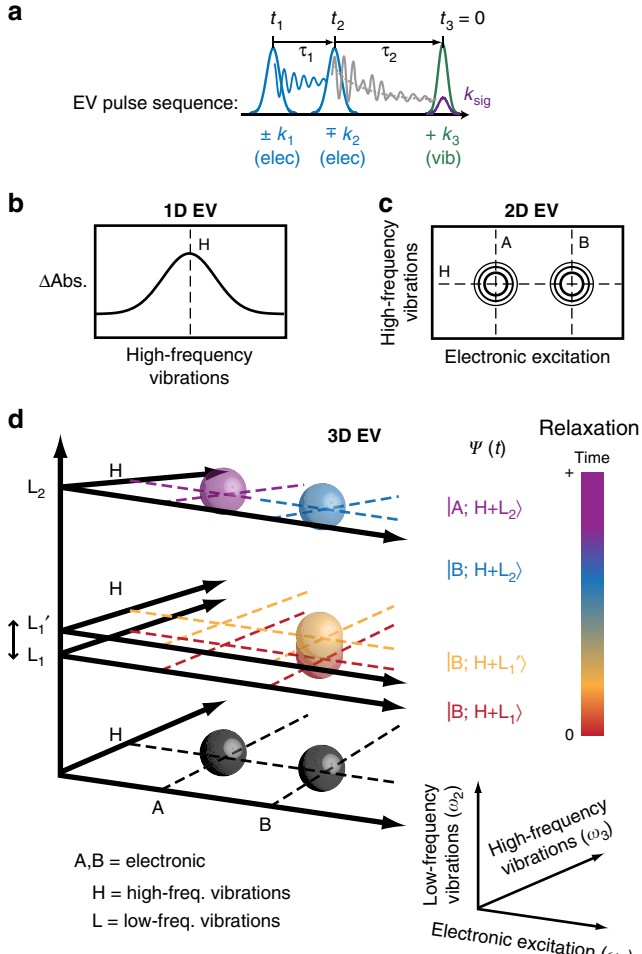

**Fig. 1** Multidimensional Electronic–Vibrational (EV) Spectroscopy. In each technique, the pulse sequence (**a**) induces an electronic excitation (pumps: **k**$_1$, **k**$_2$) and measures a high-frequency vibrational response (probe: **k**$_3$) in a sample. Consider a system where one high-frequency vibration (H) vibronically couples to two electronic excited states (A, B) and to two low-frequency vibrations (L$_1$, L$_2$) in A and/or B. **b** The pump–probe experiment (1D EV) measures the H vibrational spectrum ($\omega_3$) and kinetics during its relaxation times ($\tau_2$). At a given $\tau_2$ delay, (**c**) the 2D EV experiment resolves the electronic excitation spectrum ($\omega_1$) for H, defining the excited vibronic eigenstates ($\psi(t)$) by the couplings between H, the ground electronic state and either A or B. Collecting 2D EV spectra over a $\tau_2$ range measures population relaxation and coherent superpositions of eigenstates. The latter manifest as peak amplitude modulations; a FT over $\tau_2$ yields a (**d**) 3D EV spectrum where coherences (rainbow-colored spheres) separate in $\omega_2$ from population decays (black spheres) according to their beat frequency. The $\omega_2$ frequencies can indicate low-frequency vibrations (L$_1$, L$_2$) coupled with vibronic eigenstates. Time-dependent 3D EV features, such as frequency shifts of low-frequency modes as generally indicated by L$_1$ moving to L$_1$′ in the $\omega_2$ frequency space, report on the temporal evolution and nonequilibrium relaxation of the vibronic eigenstates during photoexcited processes (depicted by the sphere coloration gradient).

photoisomerization[40], no reported study to date has fully exploited the potential of 3D FT spectroscopy to most completely describe molecular eigenstates in terms of three energetically disparate molecular coordinates. Here, the complete capability of multidimensional EV spectroscopy is demonstrated by directly following vibronic coherence evolution in the photoexcited aqueous N3$^{4-}$ molecule (fully deprotonated N3) while it undergoes ultrafast ISC and nonadiabatic IC. An

in-depth characterization of an excited-state vibrational wave packet is reported here in photoexcited N3$^{4-}$ as it propagates during the initial 600 femtoseconds (fs), answering an outstanding question about the wave packet's existence within the N3 triplet manifold[12]. Consequent energy transfer from the initial vibrational wave packet to a second coherence is observed, which propagates further for ~1 picosecond (ps). Interestingly, the electronic character of the evolving secondary coherence oscillates between two different MLCT states with 340 ± 40 fs periodicity. Our measurement identifies this as a vibronic coherence promoting nonadiabatic IC. We simulate the experimental result and extract the 50 cm$^{-1}$ nonadiabatic coupling driving IC during the excited-state triplet relaxation. This study provides an unambiguous view of a photoexcited vibronic mechanism driving long-lived charge separation in a solar cell dye.

## Results

**The MLCT states and the high-frequency vibrational reporter**. The aqueous N3$^{4-}$ has two $^1$MLCT electronic absorption bands (Supplementary Fig. 1) centered at 20,000 cm$^{-1}$ (500 nm) and 26,880 cm$^{-1}$ (372 nm). The $^1$MLCT absorption consisting of a dense manifold of MLCT excited states results in electron density with mixed Ru-(NCS)$_2$ character shifting to a dcbpy $\pi^\star$ orbital[8,41]. The $^3$MLCT excited-state manifold is also dense, which facilitates ultrafast ISC on a timescale comparable with one or two vibrational periods of the ligand high-frequency vibrational modes[6,8,13,15]. We follow the excited-state intramolecular charge donor–acceptor dynamics vibrationally. Previously, we identified principal high-frequency vibrational signatures of the excited electronic states for both the charge donor and acceptor in the N3$^{4-}$ transient IR (tIR), or 1D EV spectra (see Supplementary Note 2, Supplementary Fig. 4, and Supplementary Table 1)[34]. The symmetric stretching carboxylate vibration ($\omega_3 = 1328$ cm$^{-1}$) in the electronic excited state, referred to as $\nu_{COO}$, is a spectrally isolated reporter of the charge-accepting dcbpy ligand and is the high-frequency mode of interest here.

Using polarization-selective 2D EV spectroscopy of N3$^{4-}$ at a single $\tau_2$ time delay, we discovered that two excited electronic states with $^1$MLCT character ($^1$MLCT$_A$ and $^1$MLCT$_B$) are vibronically coupled to charge donor and acceptor vibrations within the excited triplet manifold, likely facilitating ultrafast photoexcited charge transfer[34]. We now use 3D EV spectroscopy to measure the time-dependent modulation of the vibronic couplings between $\nu_{COO}$ and both MLCT$_A$ and MLCT$_B$ due to low-frequency vibrational modes throughout the initial 1500 fs of the photochemical reaction. Here, we omit the spin label to reflect that MLCT$_A$ and MLCT$_B$ are strongly spin-mixed states, which is expected from computational work on similar complexes[8,10] and supported in our data by a consistent $\omega_1$ peak maxima of $\nu_{COO}$ (Supplementary Note 8 and Supplementary Fig. 13). The ground-state bleach (GSB, positive signal) and excited-state absorption (ESA, negative signal) of the carboxylate region are shown in the 2D EV spectra (Fig. 2a) with the MLCT$_A$, MLCT$_B$, and $\nu_{COO}$ transitions highlighted.

**$\tau_2$-Dependent amplitude oscillations in 2D EV spectra**. Nontrivial 2D EV amplitude modulations are observed over the initial 1500 fs of the $\tau_2$ delay. The oscillatory amplitude of the 2D EV peak for $\nu_{COO}$ vibronically coupled with MLCT$_B$ is highlighted in Fig. 2a. To observe the amplitude modulations more clearly, the exponential population kinetics are subtracted to obtain the residual amplitude of the integrated 2D area ($\omega_1$, $\omega_3$) for the greatest ~10–15% signal of the 2D EV peaks measuring the $\nu_{COO}$ vibronic coupling with MLCT$_A$ and MLCT$_B$ (gray in Fig. 2b, e, respectively; see Supplementary Fig. 7 and Supplementary Table 2). In accord

with previously reported wave packet formation in N3[12], two different $\tau_2$ ranges of the residuals are temporally selected (red and blue; Fig. 2b, e), and Fourier transformed to identify the principal low-frequency components modulating the 2D EV signal within those $\tau_2$ ranges (Fig. 2b–g; Supplementary Fig. 7). The details of the FT analysis are given in Supplementary Note 3. Careful consideration of instrument noise is required, as the oscillations are weak compared with the population decay (Supplementary Fig. 5). Hence, we only consider the $\omega_2$ features most prominently above experimental noise (i.e., $\nu_{Ru-N}$ and $\nu_{Ru-bpy}$) in Fig. 2c, d, f, g. The noise floor is set by performing the identical FT analysis on the 2D EV spectral region, where the signal level is ≤5% of the maximum 2D EV signal (Supplementary Note 3 and Supplementary Fig. 6). We exclude features at $\omega_2 \geq 790$ cm$^{-1}$ given our 833 cm$^{-1}$ Nyquist sampling limit and the features approaching 0 cm$^{-1}$ (DC limit) due to imperfect population kinetics subtraction. During early relaxation times (0 < $\tau_2$ < 600 fs, Fig. 2c, f), a peak at $\omega_2 = 340$ cm$^{-1}$ is measured only for the $\nu_{COO}$ vibration of MLCT$_B$ character. The electronically-localized nature of this excited vibrational wave packet reported here reveals a stricter electronic characterization of this previously reported wave packet[12]. At later relaxation times (400 < $\tau_2$ < 1500 fs), a low-frequency vibrational mode at $\omega_2 \cong 742$ cm$^{-1}$ is clearly coupled to the $\nu_{COO}$ vibration of both MLCT$_A$ and MLCT$_B$ character. These spectra show a markedly different vibrational and electronic character of the measured coherences between the early and later relaxation periods.

Previous calculations[34] of the lowest energy triplet state vibrational spectrum of N3$^{4-}$ (Supplementary Note 9 and Supplementary Fig. 14) and resonance Raman experiments[42,43] aid us in assigning the 340 cm$^{-1}$ mode and the 742 cm$^{-1}$ mode as involving the stretching coordinate between the Ru and the dcbpy nitrogens (respectively called $\nu_{Ru-N}$ and $\nu_{Ru-bpy}$ here), with the $\nu_{Ru-bpy}$ primarily localized to the dcbpy ring. A recent surface-hopping trajectory calculation on [Ru(bpy)$_3$]$^{2+}$, a molecule similar to N3[8], included on-the-fly, spin–orbit coupling and found low-frequency vibrations analogous to $\nu_{Ru-N}$ and $\nu_{Ru-bpy}$ promoting ultrafast ISC due to excited-state mixing. To understand the ligand's role in realizing long-lived charge separation, the relevant electronic and vibrational coordinates and their couplings must be mapped out. From Fig. 2, the vibronic eigenstate basis explored here for N3$^{4-}$ is established and consists of: the excited electronic states (MLCT$_A$/MLCT$_B$), the charge-accepting high-frequency vibration ($\nu_{COO}$), and the low-frequency vibrations ($\nu_{Ru-N}$, $\nu_{Ru-bpy}$). An effective Hamiltonian for a complicated, multi-coordinate system is distilled down to these measured vibronic eigenstates. We introduce the eigenstates and a density matrix element notation for use in the remaining discussion: $\rho_{ab}(\tau_2) = |a(\tau_2)\rangle\langle b(\tau_2)|$ specifies a density matrix element, and $a$, $b \in \{1–8\}$ are the numbered vibronic eigenstates given in Fig. 3 (e.g., the $\nu_{Ru-N}$ coherence is represented: $|B; 0, 0, 0\rangle\langle B; 1, 0, 0| = \rho_{35}$).

**The early-time excited-state vibrational wave packet**. The 3D EV experiment provides a more complete description of the vibronic states involved in the coherent superposition composing the MLCT$_B$ excited-state vibrational wave packet measured at $\omega_2 = 340$ cm$^{-1}$. The red $\omega_2$ spectrum in Fig. 2f demonstrates that the 2D EV peak intensity oscillations at early times arise from the $\rho_{35}$ density matrix element. We characterize the time-dependence of the frequencies in the vibrational wave packet by the spectral phase, $\phi(\omega_2) = \tan^{-1}\left(\frac{\text{Im}[FT(S(\tau_2))]}{\text{Re}[FT(S(\tau_2))]}\right)$, where $S(\tau_2)$ is the 2D EV residual signal during 0 < $\tau_2$ < 600 fs for MLCT$_B$ shown in Fig. 2e (red). The group delay is obtained from the spectral phase, and is defined as $\tau_d(\omega_2) = -d\phi(\omega_2)/d\omega_2$. The spectral phase contains time versus

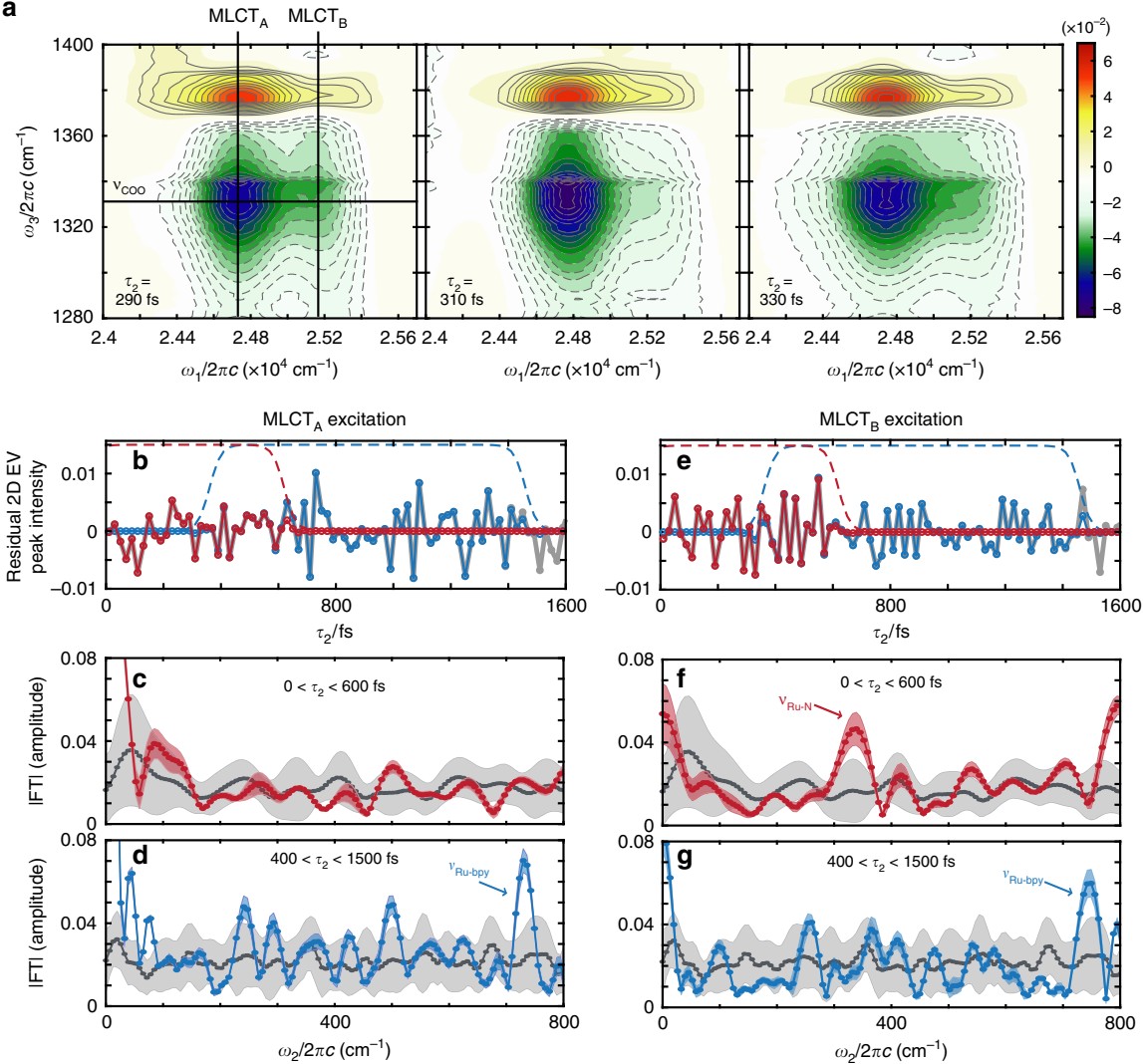

**Fig. 2** Low-frequency amplitude modulations in two-dimensional electronic–vibrational (2D EV) spectra. **a** $\tau_2$-dependent 2D EV spectra show oscillatory amplitude; notably, the $\nu_{COO}$ ($\omega_3 = 1328$ cm$^{-1}$) excited-state vibration characterized by MLCT$_B$ ($\omega_1 \cong 25,200$ cm$^{-1}$) excitation. Contour lines: solid (dashed) = positive (negative) signal, 5% intervals beginning at ± 10%. Residual intensity of the integrated ($\omega_1$, $\omega_3$) 2D peak area for the $\nu_{COO}$ coupled to the (**b**) MLCT$_A$ ($\omega_1 \cong 24,700$ cm$^{-1}$) and (**e**) MLCT$_B$ electronic excited states (gray, circles). Temporal filtering (dashed) isolates early relaxation times (0 < $\tau_2$ < 600 fs, red) or later relaxation times (400 < $\tau_2$ < 1500 fs, blue) to identify transient, low-frequency amplitude modulations in either time range. Fourier Transform (FT) spectra of the temporally filtered [(**c**) early = red; (**d**) later = blue] 2D residuals of MLCT$_A$ excitation show principal low-frequency components (i.e., $\nu_{Ru-N}$ and/or $\nu_{Ru-bpy}$) responsible for the 2D EV peak amplitude modulation. Corresponding FT spectra for MLCT$_B$ excitation are shown in (**f**) and (**g**). In all FT spectra, the gray spectra are the background (no EV signal); the shaded regions represent ±1 standard deviation from the average data (shown by circles). See Supplementary Note 3 for discussion of data processing. The number of ($\omega_1$, $\omega_3$) points: MLCT$_A$ = 45; MLCT$_B$ = 24; Background (signal ≤ 5% max) = 3295.

frequency information of a wave packet; for example, a quadratic variation in $\phi(\omega)$ corresponds to a linear group delay because the frequencies of the wave packet are changing linearly in time (see Supplementary Note 4 and Supplementary Fig. 8a, b). We note that the extracted group delay (Fig. 4, green), resulting from the best fit of the spectral phase, is positive and predominantly quadratic within the $\omega_2$ region defined by the spectral FWHM (gray arrows). Interestingly, this form of the group delay describes the wave packet shifting to higher frequencies (300 cm$^{-1}$ → 370 cm$^{-1}$ over 140 fs → 520 fs in $\tau_d$) in time. The blue-shift of the vibrational frequencies as a function of $\tau_2$ is also revealed through a sliding window short-time Fourier transform (STFT) (Supplementary Fig. 9). We attribute the change in vibrational frequencies to the

nonequilibrium relaxation of the excited wave packet with respect to the $\nu_{Ru-N}$ coordinate measured experimentally during 0 < $\tau_2$ < 600 fs. This dynamic blue-shift is suggestive of a rapid vibrational relaxation of the highly excited $\nu_{Ru-N}$ coordinate on MLCT$_B$ down the multidimensional anharmonic potential of the $\nu_{Ru-N}$. Similar observations of blue-shifting of vibrational modes on photoexcited multidimensional surfaces have been made in organic complexes[44] and in reports of bridging cyanide ligand vibrational relaxation in mixed-valence complexes during photoexcited charge transfer[45]. Consistent with earlier studies[12], the initial wave packet has largely disappeared by 600 fs as shown by the absence of an $\omega_2 = 340$ cm$^{-1}$ peak in Fig. 2d, g and it reflects the diminishing of the $\rho_{35}$ density matrix element, triggering subsequent electronic delocalization.

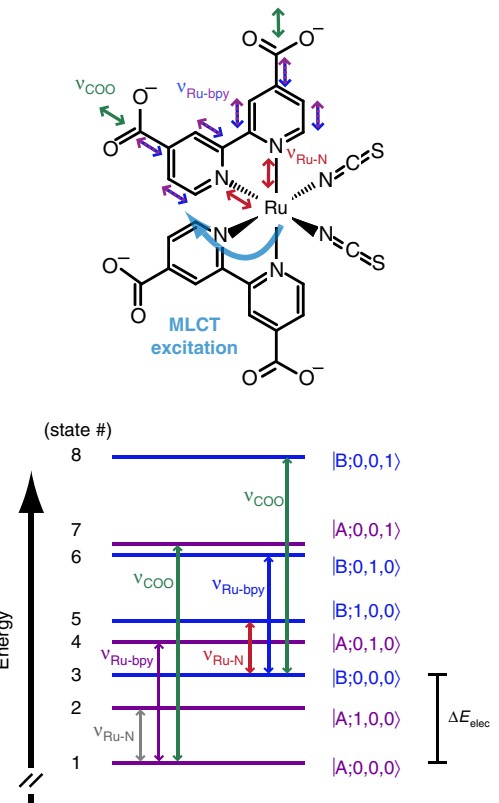

**Fig. 3** Vibronic eigenstate basis and energy-level diagram. The measured eigenstate basis of N3$^{4-}$ is cast in terms of $\left| \text{elec}; \nu_{\text{Ru-N}}, \nu_{\text{Ru-bpy}}, \nu_{\text{COO}} \right\rangle$, where elec = A (MLCT$_\text{A}$, purple) or B (MLCT$_\text{B}$, blue) and the three following quantum numbers relate the quanta in each of the vibrational modes listed in increasing transition energy. The relative energies are derived from the multidimensional EV measurements: $\Delta E_{\text{elec}} \cong 500 \text{ cm}^{-1}$ from $\omega_1$; $\nu_{\text{Ru-N}} = 340 \text{ cm}^{-1}$ and $\nu_{\text{Ru-bpy}} = 742 \text{ cm}^{-1}$ from $\omega_2$; and $\nu_{\text{COO}} = 1328 \text{ cm}^{-1}$ from $\omega_3$. The eigenstates are numbered in increasing energy for reference throughout the discussion. The N3$^{4-}$ structure (top) schematically depicts the electronic excitation (light blue) and the vibrational motions involved with photoexcited charge transfer. The vibrational mode arrow colors correspond to the respective transitions given in the energy-level diagram below.

## Vibronic coherence aids nonadiabatic internal conversion. The later time $(400 < \tau_2 < 1500 \text{ fs})$ dynamics utilize different low-frequency vibrational mode couplings with the vibronic states. The $\omega_2 \cong 742 \text{ cm}^{-1}$ features in Fig. 2d, g demonstrate that $\nu_{\text{Ru-bpy}}$ has a time-dependent coupling with $\nu_{\text{COO}}$ and participates in the $\rho_{14}$ and $\rho_{36}$ coherences. The $\nu_{\text{Ru-bpy}}$ mode also changes the Ru-N (of dcbpy) distance, emphasizing the importance of this coordinate for ultrafast relaxation in N3$^{4-}$. Since the $\omega_2$ spectra (Fig. 2) lack dynamical information occurring on shorter timescales within the $400 < \tau_2 < 1500 \text{ fs}$ period, a careful time–frequency analysis[25] (Fig. 5a,b) is used to resolve oscillatory electronic character of the coherences involving $\nu_{\text{Ru-bpy}}$. Proper selection of the temporal filter used in our analysis is required to extract the transient $\nu_{\text{Ru-bpy}}$ dynamics and reasonably interpret the result (Supplementary Note 6). Importantly, the $\omega_2$ spectra during the later time period $(400 < \tau_2 < 1500 \text{ fs}$, Fig. 2d, g) show that the data sufficiently resolve only one low-frequency mode coupling: the $\nu_{\text{Ru-bpy}}$ mode at 742 cm$^{-1}$ which has a vibrational period of 45 fs. This is a case where a reliable time–frequency analysis is obtainable with a sliding window STFT method[46]. We use a temporal window that targets the presence of 45 fs oscillations in the $\tau_2$-dependent data to understand the dynamical behavior of

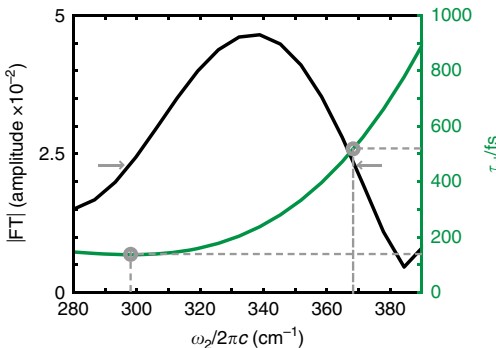

**Fig. 4** Early MLCT relaxation dynamics: electronically localized excited-state vibrational wave packet. The $\omega_2$ spectral amplitude (black; FWHM = gray arrows) of the MLCT$_\text{B}$ excited-state vibrational wave packet arises due to coherence between the $\nu_{\text{COO}}$ and one quantum of a Ru-N stretching mode ($\nu_{\text{Ru-N}} = 340 \text{ cm}^{-1}$) within the excited-state manifold. The $\tau_\text{d}$-dependence of the $\omega_2$ frequencies composing the wave packet (green) indicate blue-shifting as the wave packet propagates. Although the analytical form of $\tau_\text{d}(\omega_2)$ depends heavily on the polynomial function used to fit $\phi(\omega_2)$, the blue-shifting behavior of the wave packet is consistent across many functions. See Supplementary Note 4, Supplementary Fig. 8, and Supplementary Table 3 for spectral phase fits.

the $\nu_{\text{Ru-bpy}}$ mode (see Supplementary Note 6 and Supplementary Figs. 11, 12 for extensive discussion of temporal windowing). We emphasize that our STFT analysis provides a stand-alone interpretation of the data, because only one low-frequency mode is relevant in our later time data as identified in Fig. 2d, g, which is independent of the time–frequency analysis used in Fig. 5. It is worthwhile to note here that the STFT method becomes less suitable as multiple frequency components contribute to a time-dependent signal of interest. This is because no single temporal window can be optimal for resolving many different frequency components in a time–frequency analysis, especially if they vary significantly in frequency and transient behavior. For those complicated cases, more advanced time–frequency transforms are required for reliable interpretation[46]. From our STFT analysis, a vibronic coherence is uncovered persisting throughout this later ~1 ps of relaxation, which facilitates nonadiabatic IC.

The $\rho_{36}$ coherence appears (Fig. 5b, blue) as the initially excited wave packet diminishes. Thereafter, the $\rho_{36}$ coherence transfers to $\rho_{14}$—changing in electronic character from MLCT$_\text{B}$ to MLCT$_\text{A}$ by $\tau_2 \cong 650$–700 fs. This oscillatory dynamic continues for ~1 ps, as the amplitude of the $\rho_{36}$ and $\rho_{14}$ coherences oscillate out-of-phase with periodicity of 340 ± 40 fs. We simulate the time-dependent behavior of $\rho_{36}$ and $\rho_{14}$ as an isolated two-level system, where a 50 cm$^{-1}$ nonadiabatic coupling reproduces the 340 ± 40 fs Rabi-like oscillation seen in the experimental data (Fig. 5c; Supplementary Note 7). Here, the observed Rabi-like oscillations result from the vibronic coherence transfer between $\rho_{36}$ and $\rho_{14}$[47].

## Discussion
There is a great need for experimental methods that provide unambiguous measurement of vibronic coherences and couplings in order to uncover their role in ultrafast charge and energy transfer[5,24,33]. The results presented above clearly demonstrate the capability of multidimensional EV spectroscopy to directly monitor the temporal evolution of vibronic coherences and couplings during multi-coordinate photochemical processes. The progression of the photoexcited charge transfer process in N3$^{4-}$ measured in this study is summarized in Fig. 6.

The early-time $\rho_{35}$ coherence reveals an impressive degree of intramolecular couplings that propagate for the initial ~600 fs.

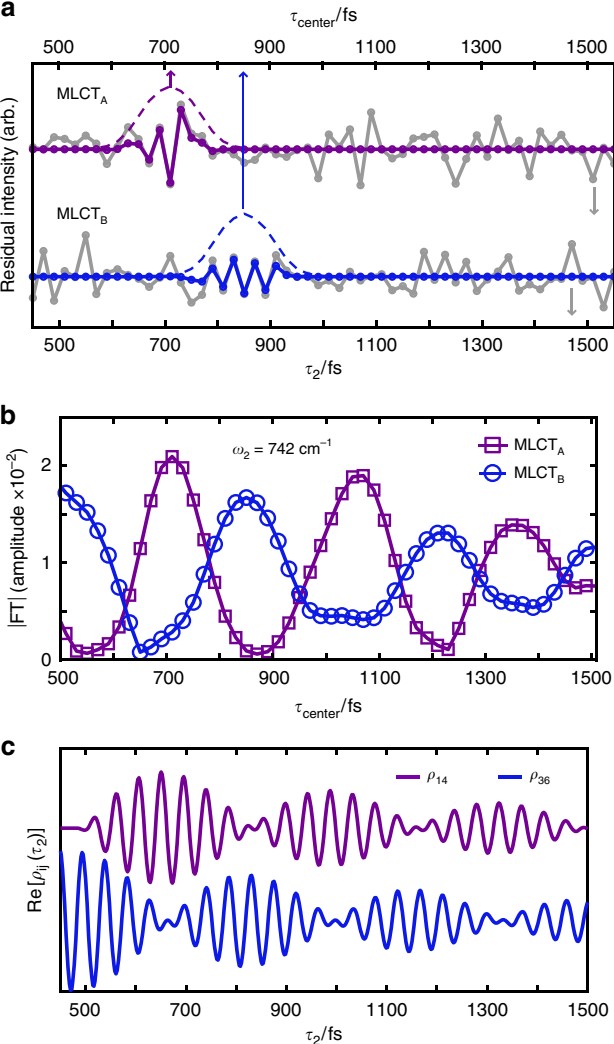

**Fig. 5** Later MLCT relaxation dynamics: nonadiabatic internal conversion through vibronic coherence. **a** Residual intensity (gray) of the integrated ($\omega_1$, $\omega_3$) 2D peak areas for the $\nu_{COO}$ vibration with MLCT$_A$ (top) and MLCT$_B$ (bottom) character; data are offset for plotting. Examples of the temporal filters (purple/blue, dashed) used in short-time FT analysis (FWHM = 120 fs, centered at $\tau_{center}$ (top axis)) of the $\tau_2$-dependent residual intensity. The purple window ($\tau_{center} = 710$ fs) highlights the $\nu_{Ru-bpy}$ (742 cm$^{-1}$) coherent beating with $\nu_{COO}$ having exclusively MLCT$_A$ character while at a later delay ($\tau_{center} = 850$ fs) the blue window shows the $\nu_{Ru-bpy}$ coherence having exclusively MLCT$_B$ character. **b** Short-time FT results reveal the oscillatory electronic character of the $\nu_{Ru-bpy}$ coherence signature between MLCT$_A$ and MLCT$_B$ with 340 ± 40 fs periodicity. **c** Simulation of the coherence block in the Redfield relaxation tensor for $\rho_{14}$(purple) and $\rho_{36}$(blue) during $\tau_2$ with an 800 fs coherence decay time where $\omega_{14} = \omega_{36} = 742$ cm$^{-1}$ and a 50 cm$^{-1}$ nonadiabatic coupling reproduces electronic oscillatory behavior for the vibronic coherence involving $\nu_{Ru-bpy}$ with MLCT$_A$ and MLCT$_B$ (data offset). See Supplementary Note 5 for discussion of Redfield relaxation dynamics and Supplementary Note 7 for simulation details.

This wave packet is confined to MLCT$_B$ electronic character, indicating that the Ru-N distance of the $\nu_{Ru-N}$ stretching coordinate specifically modulates the vibronic coupling between $\nu_{COO}$ and the MLCT$_B$ during the initial relaxation of photoexcited N3$^{4-}$ (Fig. 6b, red well). The early-time vibrational wave packet measurement clearly differentiates the nature of the vibronic coupling between either MLCT$_A$ or MLCT$_B$ and the $\nu_{COO}$, which

is a distinction that could not be made from a single 2D EV, tIR, or transient absorption (TA) experiment. The nature of the initial excited-state wave packet, first reported by Kallioinen[12] using TA, can now be confidently assigned as an excited-state vibrational coherence between $\nu_{COO}$ and $\nu_{Ru-N}$ due to the electronically localized 3D EV signal. Confirmation of the triplet character of this excited-state vibrational wave packet is clear because the signal is detected through the coherence with the triplet state $\nu_{COO}$, which answers this outstanding question in the field of N3 photophysics. We note that the known ISC time ($\leq 100$ fs) leaves the existence of the singlet vibrational wave packet ambiguous because the $\nu_{Ru-N}$ vibrational period is on the same order as ISC, if not longer.

A transition occurs between the early and later relaxation periods as the wave packet arising from the $\rho_{35}$ coherence diminishes and the later time wave packet formed by the $\rho_{36}$ coherence emerges (Fig. 6b, red to blue well). The fact that this transfer occurs between states of the same electronic character suggests that excited-state vibrational coherence transfer bridges the early-time and later time dynamics; however, many other coordinates could also affect the $\rho_{35}$ coherence. Indeed, we excite only a small window of the $^1$MLCT absorption; the observed dynamics will depend on the electronic structure of the initially excited states and the solvation environment[48,49]. Since our pump center frequency and bandwidth cannot directly excite $\rho_{36}$, this coherence must be accessed through intramolecular energy transfer. Conversely, the $\rho_{14}$ is directly excitable, yet its signature is negligible until after the $\rho_{36}$ coherence has formed (Fig. 5b). Importantly, our probe center frequency and bandwidth is insufficient to collapse the coherences observed during $\tau_2$ and excite the v = 0 → 1 $\nu_{COO}$ vibrational transition required for detecting the ESA at $\omega_3 = 1328$ cm$^{-1}$ (Supplementary Note 5 and Supplementary Table 4). Accounting for the quantum pathways capable of producing the measured oscillatory residuals implies that coherence-to-population transfer[50] plays an important role in the ultrafast photophysics of N3$^{4-}$ (see Supplementary Fig. 10). Together, these dynamics suggest that the vibronic couplings of MLCT$_B$ function as a gateway for the initially excited vibrational coherence to transition into nonadiabatic IC and continue equilibrating within the excited triplet manifold.

Discovering the out-of-phase oscillations between the $\rho_{36}$ and $\rho_{14}$ coherences is key to characterizing the wave packet evolving during the later relaxation period as an excited-state vibronic coherence (Fig. 6c). By comparison, a purely vibrational coherence, such as $\rho_{35}$, would be confined within a single electronic state. A purely electronic coherence would be expected to appear at the earliest delay times, to dephase within 10s–100s of fs, and to yield $\omega_2$ features at the difference frequency between the two electronic states independently of any vibrational coordinate. As we report here for $400 < \tau_2 < 1500$ fs, the amplitude of the vibrational coherence oscillates with the electronic character, it appears after 600 fs of relaxation, and it proceeds for ~1 ps. This prolonged coherence is characteristically vibronic, as the expected decoherence time slows when a more complete specification of the system eigenstates occurs;[29] we directly observe this quantum coherence with multidimensional EV spectroscopy. The 50 cm$^{-1}$ nonadiabatic coupling extracted from our simulations is consistent with the 340 ± 40 fs vibronic coherence transfer rate between $\rho_{36}$ and $\rho_{14}$, and the near-complete transfer observed in Fig. 5b suggests the system is within the strong coupling regime. The combination of the 2D EV experiment to map out vibronic couplings between $\nu_{COO}$ and either MLCT$_A$ or MLCT$_B$[34], and the 3D EV experiment to resolve additional excited-state coherences involving $\nu_{Ru-N}$ and $\nu_{Ru-bpy}$ allow the evolution of photoexcited vibronic coherence to be unambiguously measured.

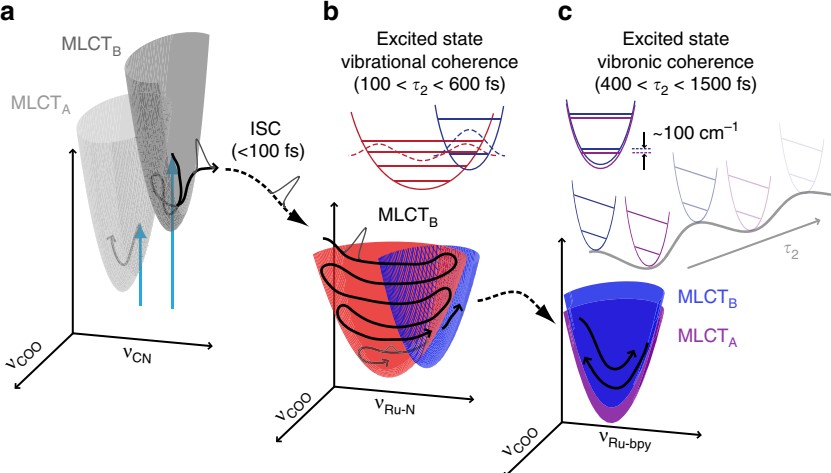

**Fig. 6** Coherence evolution during the relaxation of photoexcited N3$^{4-}$. A schematic representation of the first two picoseconds of photoexcited relaxation progresses from left to right (**a–c**). Initial MLCT electronic excitation (**a**) with MLCT states coupling the donor–acceptor vibrations shown (MLCT$_A$ = light gray well; MLCT$_B$ = dark gray well) and the initial wave packet passing through ultrafast ISC. The ~600 fs propagation of the initial excited-state vibrational coherence $\rho_{35}$ (**b**) involving $\nu_{Ru-N}$ of MLCT$_B$ (red wells); a coherence transfer to $\nu_{Ru-bpy}$ of MLCT$_B$ (blue wells) is merely suggested with sufficient wavefunction overlap involving the $\nu_{Ru-bpy}$ lowest energy vibration. The ~1 ps propagation of the excited-state vibronic coherences $\rho_{36}$ and $\rho_{14}$ involving $\nu_{Ru-bpy}$ (**c**) depicts the nonadiabatic internal conversion between MLCT$_A$ (purple) and MLCT$_B$ (blue) through the 50 cm$^{-1}$ nonadiabatic coupling strength.

Progress toward tackling the long-standing challenge of characterizing nonequilibrated charge transfer mechanisms in photoexcited donor–acceptor complexes is achieved in this study. Our 3D EV measurements provide an account of how intramolecular charge transfer occurs in an archetypal photosensitizer, detailing the complex interplay between valence electron density and molecular structural dynamics on the fs timescale. We have shown that the cascading relaxation through the nonequilibrated MLCT manifold of N3$^{4-}$ is heavily influenced by Ru-N (of dcbpy) vibrations, which effectively bridge the charge-donating Ru-(NCS)$_2$ and charge-accepting dcbpy moieties through excited-state vibronic couplings. We conclude that at least one excited-state trajectory in aqueous N3$^{4-}$ that facilitates intramolecular charge transfer is initiated through an excited-state vibrational coherence lasting for ~600 fs before transferring to a secondary coherence that is vibronic in nature and promotes nonadiabatic IC proceeding for another ~1 ps of relaxation. These results illuminate a mechanism for how solvated N3$^{4-}$ utilizes excited state, time-dependent vibronic couplings to achieve long-lived charge separation. In addition to a high density of states and strong spin–orbit coupling, we contribute evidence that structural relaxation involving metal-ligand bonding nitrogen atoms strongly influences excited-state mixing and efficient formation of long-lived $^3$MLCT states[6,8]. This mechanistic insight may be particularly useful for designing more efficient and earth-abundant light-harvesting complexes[51], and for harnessing the typically untapped potential of unthermalized photoexcited states to control photochemical efficacy[52]. Following photochemical reactivity from the perspective of specific vibrational coordinates is a unique advantage of multidimensional EV spectroscopy, which can be exploited to identify critical intramolecular coordinates governing nonequilibrium photoexcited processes. Multidimensional EV spectroscopy will be an incisive tool for understanding photoexcited energy and charge transfer mechanisms during complex, multi-coordinate photochemical processes in molecular and material systems.

## Methods

**Experimental description.** The 2D EV experiment is performed in a similar way to a tIR experiment with a pair of electronic pump pulses responsible for creating an excited state (i.e., the "bra" and "ket" interactions) followed by an IR probe pulse; see Supplementary Note 1 and Supplementary Figs. 1–3 for complete experimental details. The $\tau_2$ delay time (the typical pump–probe relaxation time) is fixed, and the delay time ($\tau_1$) between a pump pulse pair generated in an interferometer is scanned. A FT over $\tau_1$ yields the electronic excitation frequency dimension, $\omega_1$, while the high-frequency vibrational detection dimension, $\omega_3$, is measured using a spectrometer. A single 2D EV spectrum is collected at a given $\tau_2$ delay time; it describes the electronic–vibrational frequency correlations present between the electronic states excited in $\tau_1$ and the high-frequency vibrations probed in $\tau_3$. In this way, 2D EV directly measures vibronically coupled degrees of freedom. A series of 2D EV spectra collected over a range of $\tau_2$ delays maps the time-dependence of the vibronic couplings resolved in the 2D EV spectrum. As is common to all 2D spectroscopies, $\tau_2$-dependent signal amplitudes measure population dynamics and coherent superpositions that are accessible within the excitation spectral bandwidth of the pump pulses. The UV pump center frequency ($\omega_0$) and bandwidth used in these experiments are $\omega_0 = 24{,}900$ cm$^{-1}$ and full-width-at-20%-max = 1400 cm$^{-1}$; the mid-IR probe center frequency and bandwidth used here are $\omega_0 = 1375$ cm$^{-1}$, full-width-at-20%-max = 245 cm$^{-1}$. Amplitude modulations occurring at the difference frequency between the vibronic states involved in the excited $\tau_2$ coherences are measured and can be more directly resolved as residual 2D peak intensity following subtraction of signals from population dynamics (see Supplementary Note 3 for complete details). Another FT over the residual intensities during $\tau_2$ reveals a third frequency dimension, $\omega_2$, that identifies the lower frequency vibrations and molecular coherences responsible for the amplitude modulations in the 2D EV spectra. Resolving all three frequency dimensions yields a three-dimensional (3D) EV spectrum, which in this study contains no common frequencies among any of the three spectral dimensions. Importantly, this means that our 3D EV spectrum requires a high degree of different intramolecular couplings to view 3D EV signal. Moreover, low-frequency mode couplings in $\omega_2$ may be time-dependent, appearing only at certain times during various photoexcited molecular relaxation pathways. The time-dependence of the measured vibronic couplings can be ambiguously defined, and perhaps irresolvable, without a time–frequency analysis such as the sliding window short-time FT analysis used in this work (see Supplementary Note 6).

**Instrumental details.** The fundamental output of a Ti:Sapphire regenerative amplifier (Spectra Physics Spitfire XP Pro; 800 nm, 4.0 W, ~40 femtoseconds (fs), 1 kHz) is used for these studies. The broadband UV (BBUV) pump is generated by the second harmonic of a spectrally broadened portion of the 800 nm fundamental beam using a 100 μm BBO (Type I) crystal (Newlight Photonics); the spectral broadening method is detailed in Supplementary Note 1. An acousto-optic pulse shaper (Fastlite Dazzler) generates the collinear pump pulse pair ($\mathbf{k}_1$, $\mathbf{k}_2$), delivering 240 nJ/pulse focused to a 220 μm 1/e$^2$ diameter at the sample. The pump is mechanically chopped at 500 Hz to collect differential absorption spectra. The difference frequency between the signal and idler fields of a home-built optical parametric amplifier generates the mid-IR probe, which provides 480 nJ/pulse focused to a 195 μm 1/e$^2$ diameter at the sample. The isotropic signal is collected by setting the polarization angle between the pump and probe to the magic angle (54.7°) for data collection. The instrument response has subsided by

$\tau_2 = 180$–$200$ fs given by the solvent only tIR signal and the rise time of the nonresonant integrated pump–probe signal in a 250 µm Si wafer (Supplementary Fig. 2).

## Data availability

The data supporting this study are available from the corresponding author upon reasonable request.

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

## Acknowledgements

The development of 2D EV spectroscopy is supported by the National Science Foundation (NSF) (Grant no. CHE 1565759). J.D.G. is supported by the NSF GRFP, Division of Graduate Education (No. DGE-1256082).

## Author contributions

J.D.G., J.S., and M.K. designed the experiments. J.D.G. and J.S. performed the experiments. J.D.G. analyzed the data. J.D.G., J.S., and M.K. wrote the paper.

## Competing interests

The generation of the near UV pump pulse used in the experiment is the subject of a US provisional patent application. The relevant information is given below: Provisional Patent Application 62/897,032 filed 9/6/2019 Entitled: "Efficient Generation of Stable sub-20 fs 400 nm Pulses for High Order Nonlinear Spectroscopy" Inventors: Munira Khalil, James Gaynor, Joel Leger, Jason Sandwisch.
