## [Peer Review File · Nature Communications]

Reviewers' comments:

Reviewer #1 (Remarks to the Author):

This is an excellent paper demonstrating the potential of 2DEV spectroscopy for revealing the interplay of nuclear and electronic degrees of freedom in determining ultrafast dynamics.

There are only a few places where I feel the clarity of items could be improved:

1. Figure 1 and line 49: are L1 and L1' the first and second excited quantum states of the same vibration?(not clear). Please specify.
2. Line 49 question: is L1->L1' dephasing or relaxation from a higher quantum state to a lower state?
3. Lines 144-145: how does a shift to a higher frequency relate to a cooling (de-excitation?) of the Ru-N IR mode?

Reviewer #2 (Remarks to the Author):

The authors employ electronic-vibrational multidimensional spectroscopy to follow vibronic coherence evolution in a transition-metal complex that undergoes intersystem crossing and nonadiabatic internal conversion. This multidimensional spectroscopy variant, pioneered by the group of the authors, provides direct vibrational information and how it is correlated with electronic transitions. While the method has been known, in the present work it is applied to resolve relevant mechanistic steps in a charge-transfer reaction involving multiple molecular coordinates.

The paper is very well written and provides convincing explanations. Together with the extensive Supplementary Information, the details of the challenging data analysis are clearly laid out, and the conclusions are supported by data.

The signal-to-noise ration is just at the edge of what is still feasible, where numerous artefacts associated with windowed (short-time) Fourier transformation might be relevant and have led to erroneous assignments in other studies in the literature. In general, the authors do their best to analyze these effects, for example by experimenting with and comparing different types of Fourier

window functions. For example, the size of the Fourier window can have a strong effect as shown in the SI. Thus, the authors state that it is relevant to make an "informed choice over the temporal windowing function". This seems to be the most critical part of the work because the reader might get the impression that one might select the window in such a way as to reveal what one expects anyway. In other words, the manuscript would benefit from some additional comments on the extent to which the method can provide "stand-alone" interpretation. Is the generated data sufficient to allow for unique modelling of the involved processes, and are "circular arguments" avoided?

I recommend publication of the work in Nature Communcations after addressing this minor point.

Reviewer #3 (Remarks to the Author):

In this work the authors utilize 3D electronic-vibrational (EV) spectroscopy to probe excited-state couplings between electronic states and high- and low-frequency vibrations of the solar cell dye [Ru-(dcbpy)₂(NCS)₂]⁴⁺. They identify two low-frequency vibrations (340 cm⁻¹ and 742 cm⁻¹) coupled to the 1328 cm⁻¹ excited-state COO mode on two electronic excited states (MLCT_A and MLCT_B). Time-frequency analysis shows that the 340 cm⁻¹ mode appears at early times and transiently shifts in frequency within the first 0.6 ps, while the 742 cm⁻¹ mode grows in at later times (0.4-1.5 ps). Furthermore, the mode at 742 cm⁻¹ beats with a 340 fs period at both the MLCT_A and MLCT_B bands with a 180 degree phase shift between the two for ~1 ps, which was assigned to a transfer of vibrational coherence at 340 cm⁻¹ to a delocalized vibronic coherence at 742 cm⁻¹ with ~50 cm⁻¹ coupling between the two MLCT states.

The role of vibrational coherence, and particularly its relation to delocalized excited states, is a growing area of investigation because recent studies have suggested that coherent nuclear motions play a role in light harvesting and subsequent reactivity in natural and inorganic systems. Building on their previous work using 2DEV to identify couplings between electronic states and high-frequency vibrational modes that mediate charge separation, here the authors extend the experiment to a third dimension to correlate with a low-frequency vibrational mode. This allowed them to not only identify low-frequency modes that are likely relevant to ultrafast relaxation, but also identify signatures of electronic coupling on the excited state. This is a novel experimental approach that has not been demonstrated across electronic and vibrational frequency regimes. However, some experimental details and explanation critical to assess the validity of their analysis and interpretation are missing. If these are satisfactorily provided (described below), I recommend this manuscript for publication *Nature Communications*.

Major Comments:

- (1) The data shown in Figure 2 has a low signal-to-noise level. While I appreciate the authors' careful consideration of SNR in section 3 of the supplemental information, a more robust argument could be made by including time and frequency domain analysis of the tIR "pump-probe" data that, as shown, has a higher SNR. This would allow for clearer verification of the existence and time varying frequency of the 340 cm⁻¹ mode and the presence of the 742 cm⁻¹ mode at $\omega_3=1328$ cm⁻¹.
- (2) The correspondence with the calculated vibrational spectrum used to identify the 340 cm⁻¹, 742 cm⁻¹, and 1328 cm⁻¹ modes depicted in Figure 3 would be clearer if the calculated spectrum were included in the supplemental information. While the cited work from the authors (Gaynor et al. *J. Phys. Chem. Lett.* 2018, 9, 6289-6295) contains computational details, the low frequency region of the calculated spectrum is not shown in the SI.
- (3) The basis of Figure 4 would be clearer if the authors showed the data used to generate the green curve at a selected number of time points to supplement the function fitting dependent signal included in section 4 of the SI.
- (4) The experimental methods would be better described with a more thorough temporal characterization of the pump and probe pulses used in the supplemental information; the current SI simply states that "Signal within the instrument response due to pulse overlap at early times has diminished by $t_2 = 180$ -200 fs as estimated by the rise time of the non-resonant integrated pump-probe signal in a 250 um Si wafer", but does not state the extracted pulse

duration from this measurement. In addition, the authors the validity of the beating data shown in Figure 2b,c,e,f should be clarified given that the pulse overlap corresponds to 1/3 of the temporal window (0-600 fs) used for FT. For example, have the authors examined if the 340 cm^{-1} peak is robust to the presence/absence of pulse overlap?

- (5) Further explanation would make it clearer why the frequency of the Ru-N mode increases from 298 cm^{-1} to 340 cm^{-1} over 140 fs to 520 fs. It is currently unclear why cooling would cause an increase in frequency and how this frequency shift (as opposed to a damping) is a signature of dephasing.
- (6) While the 180 degree phase shift between the 340 fs mode between the two MLCT states is visible by eye, it is unclear if this is a vibrational mode that is out of phase between the two potential energy surfaces following initial relaxation, as there are several modes observed between 50 and 100 cm^{-1} in Figures 2e and 2g near the noise floor. The authors' argument would be more robust if they included the phase of each identified low frequency peak for both MLCT bands included in figures 2e and 2g, as to show that the out of phase character of this oscillatory feature is unique to electronic coupling and not strictly a vibrational mode.
- (7) In Figure 5b, the authors show oscillations in the amplitude of the frequency domain signal at 742 cm^{-1} , which they assign to a vibrational mode oscillating between electronic excited states. It is not clear why the absolute value of the oscillations decays to 1 as opposed to 0, which is generally observed as the magnitude of an oscillatory signal decays.

Minor Comments:

- (1) Line 162 should read, "the ρ_{36} coherence appears (Fig 5b, blue)..." not "the ρ_{36} coherence appears (Fig 5b, purple)..."
- (2) The y axis units in Figure 5b should read "amplitude" not "amplitdue".

RESPONSE TO REVIEWERS' COMMENTS

KEY:

Black italic: reviewer comments

Blue: authors' response to reviewer comments

Black non-italic: original manuscript text

Red: changes to original manuscript

Reviewer #1:

General Comment(s): *This is an excellent paper demonstrating the potential of 2DEV spectroscopy for revealing the interplay of nuclear and electronic degrees of freedom in determining ultrafast dynamics. There are only a few places where I feel the clarity of items could be improved:*

Response: We thank Reviewer 1 for their compliments regarding our paper and for suggesting important clarifications. This has resulted in a higher quality manuscript, as described in the following responses.

Comment 1: *Figure 1 and line 49: are L_1 and L_1' the first and second excited quantum states of the same vibration?(not clear). Please specify.*

Response: The red (L_1) and yellow (L_1') spheres in Figure 1 are meant to denote time-dependent evolution of the central frequency of a low-frequency vibrational mode during the pump-probe delay time (τ_2) as observed in a 3D EV spectrum. We use the "prime" to emphasize a time-dependent frequency shift of the L_1 mode. Since Figure 1 is a general schematic, we intentionally do not attribute a physical mechanism to the observation of the various low-frequencies during τ_2 . To clarify that this only depicts a generalized frequency shift, we have modified the last sentence of the Figure 1 caption to now read as follows on page 3 of the revised manuscript:

"Time-dependent 3D EV features, such as frequency shifts of low-frequency modes as generally indicated by L_1 moving to L_1' in the ω_2 frequency space, report on the temporal evolution and non-equilibrium relaxation of the vibronic eigenstates during photoexcited processes (depicted by the sphere coloration gradient)."

We discuss the physical mechanisms underlying the time-dependent frequency change of L_1 in the comments below (also see response to Reviewer 3, comment 5).

Comment 2: *Line 49 question: is $L_1 \rightarrow L_1'$ dephasing or relaxation from a higher quantum state to a lower state?*

Response: We thank Reviewer 1 for pointing out that our use of the word "dephasing" is misleading in the original manuscript. In the context of coherences and wavepacket dynamics, "dephasing" can carry a specific meaning describing the loss of the phase relationship between waves composing a coherent wavepacket. However, our intended use of the word "dephasing" was meant to communicate the time-dependence of a 3D EV peak observed in ω_2 . We have addressed this issue by making the following changes in the revised manuscript on lines 47-50,

“As depicted in Fig. 1d, vibrational coherences (red/yellow spheres, L_1/L_1') can be directly identified by an electronically-localized 3D EV feature and its **time-dependent behavior** (L_1 (red) $\rightarrow L_1'$ (yellow)) characterized through its ω_2 -**dependent** spectral phase.”

Please see our response to the next comment where we discuss the physical mechanisms underlying the time-dependent frequency change of L_1 .

Comment 3: Lines 144-145: how does a shift to a higher frequency relate to a cooling (de-excitation?) of the Ru-N IR mode?

Response: This comment refers to the following sentence on lines 143-145 of the original manuscript:

“Interestingly, the wavepacket shifts to higher frequencies from $298\text{ cm}^{-1} \rightarrow 368\text{ cm}^{-1}$ over $140\text{ fs} \rightarrow 520\text{ fs}$ in τ_2 which describes the wavepacket dephasing and could suggest a rapid cooling of the $\nu_{\text{Ru-N}}$ coordinate on MLCT_B .”

A blue-shift of a vibration as a function of the relaxation time can result from vibrational cooling or intramolecular vibrational relaxation on an anharmonic potential energy surface. A change in vibrational frequency can also result if the vibration is a direct reporter of the structural changes occurring in the molecule upon photoexcitation. Our measurement alone cannot unambiguously assign the physical origin of the $+70\text{ cm}^{-1}$ shift of the $\nu_{\text{Ru-N}}$ feature. We emphasize here that we only *suggest* that the observed ω_2 blue-shifting of the wavepacket could be explained by non-equilibrium relaxation of a highly excited $\nu_{\text{Ru-N}}$ mode down its anharmonic potential. Our suggestion is supported by previous reports of bridging ligand vibrations mediating ultrafast relaxation during photoinduced charge transfer reactions, where vibrational quanta as high as $n=6$ are reported to relax to $n=1-3$ on similar 300-600 fs timescales [Lynch, M. et al, J. Chem. Phys. 136, 241101 (2012)], and in small organic complexes during photoexcited *cis-trans* isomerization [Ishii, K. et al. Chem. Phys. Lett. 398, 400-406 (2004)].

We have improved and clarified our description of the results which characterize the early time dynamics of the excited state wavepacket. The sentence in Lines 143-145 of the original manuscript (given above) has been changed and two clarifying sentences have been added to now read in the revised manuscript lines 145-152:

Interestingly, the wavepacket shifts to higher frequencies from $298\text{ cm}^{-1} \rightarrow 368\text{ cm}^{-1}$ over $140\text{ fs} \rightarrow 520\text{ fs}$ in τ_2 which describes **the non-equilibrium vibrational relaxation of the excited wavepacket with respect to the $\nu_{\text{Ru-N}}$ coordinate. ~~dephasing and could~~ This dynamic blue-shift is suggestive of a rapid **vibrational relaxation cooling** of the highly excited $\nu_{\text{Ru-N}}$ coordinate on MLCT_B down the multidimensional anharmonic potential of the $\nu_{\text{Ru-N}}$. Similar observations of blue-shifting of vibrational modes on photoexcited multidimensional surfaces have been made in organic complexes⁴⁴ and in reports of bridging cyanide ligand vibrational relaxation in mixed-valence complexes during photoexcited charge transfer.⁴⁵**

The two references added to the manuscript (numbers 44-45) are the two papers cited in this response to Reviewer 1's comment.

Reviewer # 2

General Comment(s): *The authors employ electronic-vibrational multidimensional spectroscopy to follow vibronic coherence evolution in a transition-metal complex that undergoes intersystem crossing and nonadiabatic internal conversion. This multidimensional spectroscopy variant, pioneered by the group of the authors, provides direct vibrational information and how it is correlated with electronic transitions. While the method has been known, in the present work it is applied to resolve relevant mechanistic steps in a charge-transfer reaction involving multiple molecular coordinates.*

The paper is very well written and provides convincing explanations. Together with the extensive Supplementary Information, the details of the challenging data analysis are clearly laid out, and the conclusions are supported by data.

Response: We thank Reviewer 2 for their acknowledgement of the novelty of our work with respect to using 3D EV spectroscopy to coherently track multiple molecular coordinates during a charge transfer reaction. We have addressed their minor concerns about the data analysis in our responses below.

Comment 1: *The signal-to-noise ration is just at the edge of what is still feasible, where numerous artefacts associated with windowed (short-time) Fourier transformation might be relevant and have led to erroneous assignments in other studies in the literature. In general, the authors do their best to analyze [short-time Fourier transform] effects, for example by experimenting with and comparing different types of Fourier window functions. For example, the size of the Fourier window can have a strong effect as shown in the SI. Thus, the authors state that it is relevant to make an "informed choice over the temporal windowing function". This seems to be the most critical part of the work because the reader might get the impression that one might select the window in such a way as to reveal what one expects anyway. In other words, the manuscript would benefit from some additional comments on the extent to which the method can provide "stand-alone" interpretation. Is the generated data sufficient to allow for unique modelling of the involved processes, and are "circular arguments" avoided?*

I recommend publication of the work in Nature Communcations after addressing this minor point.

Response: We agree with Reviewer 2 that it is important to keep the signal-to-noise ratio in mind as we perform our analysis. For this reason, we have clearly indicated the signal-to-noise in the original manuscript in Figures 2c, d, f, and g. We also stress that we have been very conservative in deciding our noise floor in our data analysis. The noise floor is set by performing the identical FT analysis on the 2D spectral region where the signal level is $\leq 5\%$ of the maximum 2D EV signal. We have emphasized this by modifying the following text in the revised manuscript on lines 108-114:

~~"The details of the Fourier transform analysis are given in Supplementary Note 3. Careful consideration of instrument noise is required as the oscillations are weak compared to the population decay (Supplementary Fig. 3.1). Hence, we only consider the ω_2 features most prominently above experimental noise (i.e., V_{Ru-N} and V_{Ru-bpy}) in Figs. 2c-2d and 2f-2g. where the noise level is determined by the same FT analysis of 2D EV signal areas with magnitudes $\leq 5\%$ of the maximum 2D EV signal. The noise floor is set by performing the~~

identical FT analysis on the 2D EV spectral region where the signal level is $\leq 5\%$ of the maximum 2D EV signal (Supplementary 3.1 and Supplementary Fig. 3.2).”

This is the main reason that we have been sure to provide a thorough description of our analysis in the Supplementary Information (specifically, Notes 3 and 6), and why we limit our assessment of the observed dynamics to considering only the $v_{\text{Ru-N}}$ mode ($\omega_2 \cong 340 \text{ cm}^{-1}$) and the $v_{\text{Ru-bpy}}$ mode ($\omega_2 \cong 742 \text{ cm}^{-1}$), which are the signals with the greatest signal-to-noise ratios.

We thank Reviewer 2 for pointing out that our language used to discuss the time-frequency analysis using short-time Fourier transforms may inadvertently give the wrong impression to readers. In the original manuscript, we meant to convey that our initial assessment of the later time data shown in manuscript Fig. 2d and 2g enabled and informed our choice of a temporal window to properly isolate the oscillatory electronic character dynamics of the $v_{\text{Ru-bpy}}$ coherence. Of course, an assessment of the frequency components in the τ_2 -dependent signals that is independent from the time-frequency analysis is required for a sound analysis and is shown in the manuscript in Fig. 2. Thus, we completely agree that our manuscript would benefit from additional comments on the extent to which time-frequency methods provide “stand-alone” interpretations and avoid circular argumentation. We have added the following in the revised manuscript lines 164-179 to clarify this point:

“Importantly, the ω_2 spectra during the later time period ($400 < \tau_2 < 1,500 \text{ fs}$, Figs. 2d and 2g) show that the data sufficiently resolve only one low-frequency mode coupling: the $v_{\text{Ru-bpy}}$ mode at 742 cm^{-1} which has a vibrational period of 45 fs. This is a case where a reliable time-frequency analysis is obtainable with a sliding window short-time Fourier transform (STFT) method.⁴⁶ We use a temporal window that targets the presence of 45 fs oscillations in the τ_2 -dependent data to understand the dynamical behavior of the $v_{\text{Ru-bpy}}$ mode (see Supplementary Note 6 and Supplementary Figs. 6.1-6.2 for extensive discussion of temporal windowing). We emphasize that our STFT analysis provides a stand-alone interpretation of the data because only one low-frequency mode is relevant in our later time data as identified in Fig. 2d and 2g, which is independent of the time-frequency analysis used in Fig. 5. It is worthwhile to note here that the STFT method becomes less suitable as multiple frequency components contribute to a time-dependent signal of interest. This is because no single temporal window can be optimal for resolving many different frequency components in a time-frequency analysis, especially if they vary significantly in frequency and transient behavior. For those complicated cases, more advanced time-frequency transforms are required for reliable interpretation.⁴⁶ From our STFT analysis, a vibronic coherence is uncovered persisting throughout this later $\sim 1 \text{ ps}$ of relaxation which facilitates nonadiabatic IC.”

The reference, number 46 [Volpato, A. et al. Opt. Exp. 23, 20040 (2015)], has been added, and referenced heavily, to support the additions to our manuscript here. This paper systematically treats different time-frequency transforms as they can be applied to analyzing increasingly complex time-dependent signals in coherent 2D spectroscopy.

Furthermore, we have removed the misleading language from the Supplemental Information that was highlighted by Reviewer 2 and made the following changes to the first paragraph in the Supplemental Note 6 on pages SI19-SI20 which now reads as:

“In the present analyses, $\omega_2=742\text{ cm}^{-1}$ is selected from the ω_2 dimension of the time-frequency plot for consideration as it corresponds to the vibrational mode of interest, $\nu_{\text{Ru-bpy}}$, as identified by the ω_2 spectra shown in manuscript Fig. 2d and 2g obtained by considering all of the τ_2 -dependent data in the later time period. As described below, we can reliably use a temporal filter to isolate the $\nu_{\text{Ru-bpy}}$ dynamics because this is the only sufficiently resolved mode in the ω_2 spectra for both MLCT_A and MLCT_B during later times. ~~A most important factor in the STFT analysis is an informed choice over the temporal windowing function to appropriately filter the information of interest within a time-domain signal.~~”

Reviewer #3

General Comments: *In this work the authors utilize 3D electronic-vibrational (EV) spectroscopy to probe excited-state couplings between electronic states and high- and low-frequency vibrations of the solar cell dye $[\text{Ru}-(\text{dcbpy})_2(\text{NCS})_2]^+$. They identify two low-frequency vibrations (340 cm^{-1} and 742 cm^{-1}) coupled to the 1328 cm^{-1} excited-state COO mode on two electronic excited states (MLCT_A and MLCT_B). Time-frequency analysis shows that the 340 cm^{-1} mode appears at early times and transiently shifts in frequency within the first 0.6 ps, while the 742 cm^{-1} mode grows in at later times (0.4-1.5 ps). Furthermore, the mode at 742 cm^{-1} beats with a 340 fs period at both the MLCT_A and MLCT_B bands with a 180 degree phase shift between the two for ~ 1 ps, which was assigned to a transfer of vibrational coherence at 340 cm^{-1} to a delocalized vibronic coherence at 742 cm^{-1} with $\sim 50\text{ cm}^{-1}$ coupling between the two MLCT states.*

The role of vibrational coherence, and particularly its relation to delocalized excited states, is a growing area of investigation because recent studies have suggested that coherent nuclear motions play a role in light harvesting and subsequent reactivity in natural and inorganic systems. Building on their previous work using 2DEV to identify couplings between electronic states and high-frequency vibrational modes that mediate charge separation, here the authors extend the experiment to a third dimension to correlate with a low-frequency vibrational mode. This allowed them to not only identify low-frequency modes that are likely relevant to ultrafast relaxation, but also identify signatures of electronic coupling on the excited state. This a novel experimental approach that has not been demonstrated across electronic and vibrational frequency regimes.

Response: We thank Reviewer 3 for their acknowledgement of the novelty of our work with respect to using 3D EV spectroscopy to coherently track multiple molecular coordinates across a wide span of system frequencies. We have addressed their major and minor concerns about the data analysis and interpretation in our responses below.

Comment 1: *The data shown in Figure 2 has a low signal-to-noise level. While I appreciate the authors' careful consideration of SNR in section 3 of the supplemental information, a more robust argument could be made by including time and frequency domain analysis of the tIR “pump-probe” data that, as shown, has a higher SNR. This would allow for clearer verification of the existence and time varying frequency of the 340 cm^{-1} mode and the presence of the 742 cm^{-1} mode at $\omega_3=1328\text{ cm}^{-1}$.*

Response: Please see our responses to Reviewer 2, comment 1 regarding our assessment of signal-to-noise.

The reviewer suggests a FT analysis of the tIR data. We do not think this is a fruitful exercise for the following reasons:

- Similar to all 2D spectroscopies, the tIR is a 1D projection of the 2D EV spectrum along ω_1 (Fig. 2a). This means that small oscillations as a function of τ_2 are better analyzed using specific regions of the 2D spectrum instead of the 1D projection (tIR). This has been used very effectively and extensively in 2D IR and 2D ES spectroscopies. The reader can refer to the following examples: (i) Khalil, M.; Demirdoven, N.; Tokmakoff, A., *J. Chem. Phys.* (2004) 121, 362-73 for a 2D IR study and (ii) Engel, G. S.; Calhoun, T. R.; Read, E. L.; Ahn, T. K.; Mancal, T.; Cheng, Y. C.; Blankenship, R. E.; Fleming, G. R., *Nature* (2007) 446, 782-6 for a 2D ES study.
- For our specific case, since the 340 cm^{-1} mode is localized to only MLCT_B , it is difficult to see these oscillations in a tIR spectrum which integrates over the entire ω_1 spectral region.
- Similarly, the 742 cm^{-1} mode is out-of-phase on MLCT_A and MLCT_B resulting in 742 cm^{-1} oscillations only ever occurring from one MLCT state at a given time during τ_2 (see manuscript Figure 5a). If we integrate over ω_1 (as is the case for the tIR experiment), it is more difficult to view these oscillations significantly above the integrated noise.

For the above reasons, the 3D EV measurement is the experiment of choice for extracting low frequency modes coupled to specific electronic states and high-frequency modes.

Comment 2: *The correspondence with the calculated vibrational spectrum used to identify the 340 cm^{-1} , 742 cm^{-1} , and 1328 cm^{-1} modes depicted in Figure 3 would be clearer if the calculated spectrum were included in the supplemental information. While the cited work from the authors (Gaynor et al. *J. Phys. Chem. Lett.* 2018, 9, 6289-6295) contains computational details, the low frequency region of the calculated spectrum is not shown in the SI.*

Response: We have reproduced the calculated vibrational spectrum reference above (from Gaynor et al. *J. Phys. Chem. Lett.* (2018) 9, 6289-6295) with an expanded view of the low-frequency region in the Supplementary Note 9. The normal mode displacements are given to exemplify modes with character of the 340 cm^{-1} and 742 cm^{-1} vibrations discussed in the manuscript. We note the 1328 cm^{-1} normal mode is already shown in the Supporting Information of the referenced paper (Ref. 23 in the revised manuscript).

The below figure and text has been added to the revised Supplementary Note 9:

Supplementary Note 9. Calculated Low-Frequency Vibrational Modes

The calculated IR spectrum of the $\text{N}3^+$ lowest energy triplet, T_0 , is reproduced from Gaynor et al.¹ to display the full spectrum, including the low-frequency region (see zoom-in of the inset in Supplementary Figure 9.1a below). The normal mode vibrations with vector displacements provide examples of the several calculated modes in the $\nu_{\text{Ru-N}}$ ($260\text{-}430\text{ cm}^{-1}$) region and the $\nu_{\text{Ru-bpy}}$ ($710\text{-}755\text{ cm}^{-1}$) region with the respective vibrational character of the low-frequency vibrations discussed in the main manuscript. Computational details for these spectra can be found in Gaynor et al.¹ and its Supporting Information.

Supplementary Figure 9.1 (a) Calculated IR spectrum of the lowest energy triplet of $N3^{4+}$ with implicit solvent; inset shows a zoom-in of dashed gray spectral region. (b) Example normal modes from calculations displaying vibrational character of the v_{Ru-N} and v_{Ru-bpy} modes. Mode 39 (303 cm^{-1}) shown for v_{Ru-N} and mode 67 (710 cm^{-1}) shown for v_{Ru-bpy} .

Corresponding changes have been made on lines 123-126 of the revised manuscript:

“Previous calculations²³ of the lowest energy triplet state vibrational spectrum of $N3^{4+}$ (Supplementary Note 9 and Supplementary Fig. 9.1) and resonance Raman experiments^{42,43} aid us in assigning the 340 cm^{-1} mode and the 742 cm^{-1} mode as involving the stretching coordinate between the Ru and the dcbpy nitrogens N (respectively called v_{Ru-N} and v_{Ru-bpy} here)...”

Comment 3: The basis of Figure 4 would be clearer if the authors showed the data used to generate the green curve at a selected number of time points to supplement the function fitting dependent signal included in section 4 of the SI.

Response: The green curve in Figure 4 was obtained from numerical differentiation of a fit to the spectral phase ($\phi(\omega)$) of experimental data as implied by the in-line equation in line 142 of the original manuscript and as described in the original Supplementary Note 4. We have added the experimental data used to obtain the fits of $\phi(\omega_2)$ to Supplementary Figure 4.1 in the revised Supplemental Information. This is the experimental data that is used to obtain the green curve shown in manuscript Figure 4. To add the experimental $\phi(\omega_2)$ data to the manuscript Figure 4 would require a third Y-axis and complicate the interpretation of the figure. We note that this same information is already reflected in the last column of Supplementary Table 4.1 through the goodness of fit figures of merit.

To further clarify the physical basis of, and how we obtain, the group delay $\tau(\omega)$ (i.e., the green curve in manuscript Figure 4), we have modified the following sentence² in the revised Supplemental Note 4:

“A meaningful description of the wavepacket ~~propagation dephasing~~ is obtained by observing the consistencies in $\tau(\omega)$, ~~obtained by numerical differentiation of for different orders of~~ the polynomial function used in the fitting ~~for different polynomial orders~~. The ~~quartic polynomial (n=4) fit shown in Supplementary Figure 4e is used to obtain the group delay curve presented in manuscript Figure 4. The green curve in manuscript Figure 4 is identical to the red curve shown in Supplementary Figure 4f.~~”

The Supplemental Figure 4.1 caption has been revised to reflect the changes made in the Figures:

Supplementary Figure 4.1 Characterizing the Initially Excited Vibrational Wavepacket. The blue lines in all figures are the same ω_2 spectra for the V_{RU-N} wavepacket ($\omega_{2,3} = 340 \text{ cm}^{-1}$) present during $0 \leq \tau_2 \leq 600 \text{ fs}$. The figures in the left column (a,c,e) plot the spectral phase, $\phi(\omega_2)$, in red; ~~experimental data is shown by the open circles and the solid lines are fits~~. The figures in the right column (b,d,f) plot the group delay, $\tau_2(\omega_2)$, in red, ~~obtained by numerical differentiation of $\phi(\omega_2)$ as explained in equation 5 of the Supplemental Information~~. The different rows denote a different order of polynomial used to fit $\phi(\omega_2)$ during the analysis.

Comment 4: *The experimental methods would be better described with a more thorough temporal characterization of the pump and probe pulses used in the supplemental information; the current SI simply states that “Signal within the instrument response due to pulse overlap at early times has diminished by $t_2 = 180\text{-}200 \text{ fs}$ as estimated by the rise time of the non-resonant integrated pump-probe signal in a 250 um Si wafer ”, but does not state the extracted pulse duration from this measurement. In addition, the authors the validity of the beating data shown in Figure 2b,c,e,f should be clarified given that the pulse overlap corresponds to 1/3 of the temporal window (0-600 fs) used for FT. For example, have the authors examined if the 340 cm^{-1} peak is robust to the presence/absence of pulse overlap?*

Response: We have reported the instrumental response time because this is the most relevant information given the scope and data analysis methods used in this paper.

- The below Figure has been added to Supplementary Note 1 to show the instrument response characterization using the non-resonant response of the silicon wafer and the non-resonant tIR signal from solvent-only at $\omega_3 = 1328 \text{ cm}^{-1}$. The decay of the instrument response by $\tau_2 > 200 \text{ fs}$ is consistent with the rise of molecular signal from N3^{4-} shown by the tIR trace at $\omega_3 = 1328 \text{ cm}^{-1}$. Supplementary Figure 1.2 is referenced in the revised manuscript line 267 in the Methods section.

Supplementary Figure 1.2 The instrument response function is assessed using the non-resonant tIR signal of a $250 \mu\text{m}$ Si wafer (black). The N3^{4-} molecular response is observed outside of pulse overlap for $\tau_2 > 200 \text{ fs}$ from the transient-IR trace for the v_{COO} excited state absorption ($\omega_3 = 1328 \text{ cm}^{-1}$, red). The signals are normalized by the absolute value of the greatest magnitude signal and overlaid for comparison. The non-resonant solvent-only tIR signal (blue; circles are data and line is three-point moving average) is scaled and offset such that zero signal is $\Delta T/T = -1$ for comparison; the solvent response diminishes for $\tau_2 > 200 \text{ fs}$, consistent with the decay of the Si signal and the onset of the N3^{4-} molecular signal.

- The following text has also been added to the revised manuscripts on Lines 303-305: “The instrument response has subsided by $\tau_2 = 180\text{-}200 \text{ fs}$ given by the solvent only tIR signal and the rise time of the non-resonant integrated pump-probe signal in a $250 \mu\text{m}$ Si wafer (Supplementary Figure 1.2).”
- Any clarification that a reader needs regarding the validity of the beating data shown in manuscript Figure 2b,c,e, and f is presented clearly in Figure 4 of the manuscript by the group delay curve shown in green. Indeed, the 340 cm^{-1} feature in ω_2 is robust to the presence and absence of pulse overlap because the group delay curve begins within pulse overlap ($\tau_2 < 200 \text{ fs}$) and extends beyond pulse overlap ($\tau_2 > 200 \text{ fs}$).

Comment 5: Further explanation would make it clearer why the frequency of the Ru-N mode increases from 298 cm^{-1} to 340 cm^{-1} over 140 fs to 520 fs . It is currently unclear why cooling would cause an increase in frequency and how this frequency shift (as opposed to a damping) is a signature of dephasing.

Response: See our response to Reviewer 1, comment 3, above.

Reviewer 3's comment has also pointed out that our intended general use of the word "dephasing" has resulted in some confusion, similar to Reviewer 1 (comment 2). To completely address comment 5 by Reviewer 3 here, we have removed the remaining uses of "dephasing" in the original main text (lines 16, 163, and Figure 4 caption).

- Lines 14-16 of revised manuscript have been edited:

"We discover an excited state vibronic mechanism driving long-lived charge separation consisting of an initial electronically-localized vibrational wavepacket which triggers delocalization onto two charge transfer states after ~~propagating for dephasing in~~ ~600 femtoseconds."

- Lines 180-181 of the revised manuscript now read:

"The ρ_{36} coherence appears (Fig. 5b, ~~blue-purple~~) as the initially excited wavepacket reaches 368 cm^{-1} by $\tau_2 = 520 \text{ fs}$ ~~during its dephasing.~~"

where the parenthetical reference to Fig. 5b has also been changed in response to Reviewer 3's minor comment below.

- The Figure 4 caption of the revised manuscript now reads:

Figure 4: Early ³MLCT relaxation dynamics: electronically-localized excited state vibrational wavepacket ~~dephasing~~. The ω_2 spectral amplitude (black; FWHM = gray arrows) of the MLCT_B excited state vibrational wavepacket arises due to coherence between the ν_{COO} and one quantum of a Ru-N stretching mode ($\nu_{\text{Ru-N}} = 340 \text{ cm}^{-1}$) within the excited ~~triplet state~~ manifold. The τ_2 -dependence of the ω_2 frequencies composing the wavepacket (green) indicate blue-shifting as the wavepacket ~~propagates-dephases~~.

Although the analytical form of $\tau_2(\omega_2)$ depends heavily on the polynomial function used to fit $\phi(\omega_2)$, the blue-shifting behavior of the wavepacket is consistent across many functions. See Supplementary ~~Note section~~ 4, ~~Supplementary Fig. 4.1~~ and ~~Supplementary Table 4.1~~ for spectral phase fits."

Comment 6: *While the 180 degree phase shift between the 340 fs mode between the two MLCT states is visible by eye, it is unclear if this is a vibrational mode that is out of phase between the two potential energy surfaces following initial relaxation, as there are several modes observed between 50 and 100 cm^{-1} in Figures 2e and 2g near the noise floor. The authors' argument would be more robust if they included the phase of each identified low frequency peak for both MLCT bands included in figures 2e and 2g, as to show that the out of phase character of this oscillatory feature is unique to electronic coupling and not strictly a vibrational mode.*

Response: We agree with Reviewer 3 that it is possible for a vibrational mode (e.g., a solvent/bath mode) to be out of phase between two potential energy surfaces and contribute to similar observations as those reported in our paper. We do not include the phase of each identified low-frequency mode (other than the 742 cm^{-1} mode which is clear as noted by the reviewer above)

for both MLCT bands included in Figs. 2d and 2g (referred to as “Figure 2e” by Reviewer 3) for the following reasons:

- As discussed in our response to Reviewer 2, we only consider modes above the noise floor designated in Fig. 2d and 2g. As stated in the original manuscript on lines 117-118 (revised manuscript lines 118-120): “At later relaxation times ($400 < \tau_2 < 1,500$ fs) a low-frequency vibrational mode at $\omega_2 \cong 742$ cm^{-1} is clearly coupled to the ν_{COO} vibration of both MLCT_A and MLCT_B character.”
- Reviewer 3 refers to the modes between 50 and 100 cm^{-1} “near the noise floor”. We note that these modes are only above the noise in Fig. 2d. We do not analyze these modes for 2 reasons: (i) given that we are interested in looking at the relative phases of these modes on the two MLCT states, we should only consider modes which are above the noise and present on both states (displayed on Figs. 2d and 2g) and (ii) as stated in the original manuscript on lines 112-114 (revised manuscript lines 114-115) : “We exclude features at $\omega_2 \geq 790$ cm^{-1} given our 833 cm^{-1} Nyquist sampling limit and the features approaching 0 cm^{-1} (DC limit) due to imperfect population kinetics subtraction.”

We emphasize that the 742 cm^{-1} mode is unique in Figure 2d and Figure 2g of the original manuscript because it is the only mode that is significant for *both* MLCT bands. This is another strong reason for limiting our analysis to this mode’s dynamics.

Comment 7: *In Figure 5b, the authors show oscillations in the amplitude of the frequency domain signal at 742 cm^{-1} , which they assign to a vibrational mode oscillating between electronic excited states. It is not clear why the absolute value of the oscillations decays to 1 as opposed to 0, which is generally observed as the magnitude of an oscillatory signal decays.*

Response: We point out that the Reviewer has perhaps misread the Y-axis of Figure 5b in manuscript, which only spans from 0 to 0.025. There are no “decays to 1” as suggested by the Reviewer. Addressing the more general comment by Reviewer 3: there is no reason why this oscillation should decay to zero because other population transfer mechanisms – both coherent and incoherent – and photoexcited relaxation mechanisms can still contribute to populating the observed states. There are many other degrees of freedom in the sample that our particular experiment is not directly sensitive to which can still effectively populate various triplet states during photoexcited relaxation. The reported experiment is sensitive to those dynamics observed within the frequency ranges that we access which are stated clearly on lines 43-45 of the original and revised manuscripts. An in-depth discussion of the transitions accessible in our study is provided in Supplemental Note 5 which is referenced in lines 196-201 of the original manuscript (lines 214-219 in revised manuscript).

Minor Comment 1: *Line 162 should read, “the ρ_{36} coherence appears (Fig 5b, blue)...” not “the ρ_{36} coherence appears (Fig 5b, purple)...”*

Response: This has been changed in the revised manuscript on Line 180. (see response to Reviewer 3 comment 5 above)

Minor Comment 2: *The y axis units in Figure 5b should read “amplitude” not “amplitdue”.*

Response: Thank you, this has been changed in Figure 5b of the revised manuscript.

Reviewers' comments:

Reviewer #2 (Remarks to the Author):

In my opinion, all issues raised by myself and the other reviewers have been addressed satisfactorily in this revised version. I gladly recommend publication in the present form.

Tobias Brixner

Reviewer #3 (Remarks to the Author):

The resubmitted manuscript has been improved in both content and clarity and we thank the authors for careful consideration of our comments. However, with this improved clarity in the resubmitted manuscript, I have an additional major comment to be addressed before publication.

The authors have improved the description of the green curve (τ_2 as a function of ω_2) in Figure 4, however with this the authors appear to be drawing several comparisons between τ_2 , the time delay between pulses 2 and 3 as depicted in Figure 1, and τ_2 , the group delay recovered from the spectral phase as described mathematically in lines 143-144, that requires further justification, as outlined in the subpoints below.

a) The nomenclature used to describe the time delay between pulses 2 and 3 and the group delay should be distinguished, as currently they are both denoted as τ_2 , consistently throughout the text and figures.

b) In assigning physical phenomena to the blueshift in frequency, ω_2 , with group delay, τ_2 , the authors cite previous work (ref. 44-45). However, these references describe a blueshift of the wavepacket as a function of time delay, τ_2 . There is no clear reason as to why the frequency dependent response of these two time parameters should be the same, especially as the group delay, τ_2 , is determined from the Fourier transform of only one time window of time delay, τ_2 . As such, the authors should justify why the group delay trends observed should correspond to a dampening effect, and if relevant, how this corresponds to relaxation timescale (Rev. 1 rebuttal point 3, Rev 3 rebuttal point 5). This assignment is in conflict with previous work, specifically work by Nabekawa et al. (DOI: 10.1038/ncomms9197) that attributed timescales of group delay to a vibrational wavepacket settling time on ~ 1 fs timescale before undergoing coherent motion.

RESPONSE TO REVIEWERS' COMMENTS DATED OCTOBER 15, 2019

KEY:

Black italic: reviewer comments

Blue: authors' response to reviewer comments

Black non-italic: original manuscript text

Red: changes to original manuscript

Reviewer #1:

From Editor: *Reviewer #1 provided comments to the editor only, in which they say they are satisfied with the responses provided.*

Reviewer # 2

In my opinion, all issues raised by myself and the other reviewers have been addressed satisfactorily in this revised version. I gladly recommend publication in the present form.

Tobias Brixner

Reviewer #3

General Comments: *The resubmitted manuscript has been improved in both content and clarity and we thank the authors for careful consideration of our comments. However, with this improved clarity in the resubmitted manuscript, I have an additional major comment to be addressed before publication.*

The authors have improved the description of the green curve (τ_2 as a function of ω_2) in Figure 4, however with this the authors appear to be drawing several comparisons between τ_2 , the time delay between pulses 2 and 3 as depicted in Figure 1, and τ_2 , the group delay recovered from the spectral phase as described mathematically in lines 143-144, that requires further justification, as outlined in the subpoints below.

Response: We thank the Reviewer for clarifying their question about Figure 4 raised earlier in the review dated August 29, 2019. We address the additional comment raised by Reviewer 3 in detail below. We have made changes to the main text and the supplementary information to distinguish between the experimental time delay, τ_2 and the group delay defined in lines 143-145 of the main text.

Comment 1a: *The nomenclature used to describe the time delay between pulses 2 and 3 and the group delay should be distinguished, as currently they are both denoted as τ_2 , consistently throughout the text and figures.*

Response: We agree that the nomenclature is confusing and have corrected it. The experimental time delay between experimental pulses 2 and 3 is referred to as τ_2 throughout the text and figures. The group delay, defined as the derivative of the spectral phase ($\varphi(\omega_2)$), is now denoted as $\tau_\sigma(\omega_2)$ throughout the text. Note that ω_2 refers to the frequency obtained after a Fourier Transform of the data from $0 < \tau_2 < 600$ fs (see Figures 2e and 2f). Below is a list of changes made to replace τ_2 with $\tau_\sigma(\omega_2)$ and more clearly distinguish them when discussing the group delay:

- i. Various instances in Lines 144-151 (see response to Comment 1b below).
- ii. Figure 4 and Figure 4 caption.
- iii. Line 149

- iv. Lines 188-189
- v. Line 218
- vi. Supplementary Note 4.
- vii. Supplementary Figure 4.1 (b), (d) and (f) and corresponding Figure Caption.

Comment 1 b: *In assigning physical phenomena to the blueshift in frequency, ω_2 , with group delay, τ_2 , the authors cite previous work (ref. 44-45). However, these references describe a blueshift of the wavepacket as a function of time delay, τ_2 . There is no clear reason as to why the frequency dependent response of these two time parameters should be the same, especially as the group delay, τ_2 , is determined from the Fourier transform of only one time window of time delay, τ_2 . As such, the authors should justify why the group delay trends observed should correspond to a dampening effect, and if relevant, how this corresponds to relaxation timescale (Rev. 1 rebuttal point 3, Rev 3 rebuttal point 5). This assignment is in conflict with previous work, specifically work by Nabekawa et al. (DOI: 10.1038/ncomms9197) that attributed timescales of group delay to a vibrational wavepacket settling time on ~ 1 fs timescale before undergoing coherent motion.*

Response: There are several points to be addressed in response to this comment and we will break up our discussion with separate bullet points below.

- The first concern raised is that since the extracted group delay, τ_d , is not the same as the experimental delay τ_2 , the frequency response of the two variables do not necessarily have to follow the same trend. We agree with this concern and we note that our use of different variables for the group delay and experimental time delay alleviates some of the confusion in the revised text on Lines 139 – 162. If the spectral phase varies with frequency, it indicates that the wavepacket experiences variation in frequency with time. The group delay is a measure of the variation of the spectral phase with frequency. In Figure 4, we see a positive change of the group delay with frequency. This result indicates that the bluer frequencies in the data arrive later in time during the experiment. Since τ_d is not the inverse of ω_2 , we cannot say *when* in the experimental time window of 0-600 fs the frequencies are blue-shifting.
- We could have chosen to perform a sliding window short time Fourier transform (STFT) analysis of the early time data to show the blue-shifting of the frequency. We have added this analysis in Supplementary Note 4 and Supplementary Fig. 4.2. This figure clearly shows the blue-shifting of the vibrational frequencies by ~ 60 cm^{-1} when τ_{center} changes from 250 – 570 fs. This analysis confirms our conclusion that positive slope of the group delay as a function of frequency reflects the physical phenomena of the blue-shifting of the wave-packet as a function of τ_2 . The reason we decided to use the spectral phase/group delay analysis method to highlight in the main text is to avoid the pitfalls of the STFT method when multiple frequency components contribute to a time-dependent signal. We have discussed this in detail on Lines 170-185 in the revised main text and in Supplementary Note 6 and Supplementary Figures. 6.1 and 6.2.
- In light of the points raised above, we are justified in assigning the extracted group delay to the blue-shifting of the wavepacket and attributing it to the cooling of the $v_{\text{Ru-N}}$ coordinate on an anharmonic multidimensional potential. As noted in the main text, our observation is similar to previously cited work in Refs. 44 and 45. The main difference being that the analysis was performed differently in those cases. The analysis of the spectral phase and the group delay shown in our work is an alternative way to characterize the time-dependent frequency change in an ultrafast pump-probe or multidimensional experiment (similar to characterizing an ultrafast laser pulse).

- The reviewer asks us to justify “*why the group delay trends observed should correspond to a dampening effect, and if relevant, how this corresponds to relaxation timescale (Rev. 1 rebuttal point 3, Rev 3 rebuttal point 5).*” We note that following our first revision, we do not associate the group delay with the decay time of the wavepacket. As stated above, the group delay reflects the blue-shifting of the frequencies with time. With regards to the damping of the amplitude of the wavepacket, we can only say that it is absent at $\tau_2 > 600$ fs. This is shown by the absence of the ν_{Ru-N} vibration above the noise floor in Figures 2d and 2g obtained after a FT of the data for later times, $400 < \tau_2 < 1,500$ fs.
- Finally, Reviewer 3 brings up the “*work by Nabekawa et al. (DOI: 10.1038/ncomms9197) that attributed timescales of group delay to a vibrational wavepacket settling time on ~ 1 fs timescale before undergoing coherent motion.*” We do not see any connection between the work cited above and our manuscript. Specifically, the Nabekawa paper mentioned above and an earlier paper (DOI: 10.1038/srep11366) describe a gas phase, ionization of a diatomic molecule. We are describing the formation of a wavepacket on a multidimensional MLCT excited state of a large transition metal complex dissolved in solution. The vast differences in the size and phase of the molecular system and the nature of the ionized vs. a MLCT excited state, do not allow us to make a useful comparison of the two studies. For these reasons, we do not discuss this further.
- We have addressed all of the above points and made modifications to the main text on Lines 139 – 162 of the revised text, which now read:

“The 3D EV experiment provides a more complete description of the vibronic states involved in the coherent superposition composing the $MLCT_B$ excited state vibrational wavepacket measured at $\omega_2 = 340 \text{ cm}^{-1}$. The red ω_2 spectrum in Fig. 2f demonstrates that the 2D EV peak intensity oscillations at early times arise from the ρ_{35} density matrix element. We characterize the time-dependence of the frequencies in ~~The relaxation of~~ the vibrational wavepacket ~~on a vibronic state~~

~~may be is characterized~~ by ~~it’s the~~ spectral phase, $\phi(\omega_2) = \tan^{-1} \left(\frac{\text{Im}[FT(\tau_2)]}{\text{Re}[FT(\tau_2)]} \right)$, and group

delay, $\tau_d(\omega_2) = -\frac{d\phi(\omega_2)}{d\omega_2}$. The spectral phase contains time versus frequency information

of a wavepacket; for example, a quadratic variation in $\phi(\omega)$ corresponds to a linear group delay because the frequencies of the wavepacket are changing linearly in time (see Supplementary Note 4 and Supplementary Figures 4.1a and 4.1b). We note that the extracted group delay (Fig. 4, green), resulting from the best fit of the spectral phase, is positive and predominantly quadratic within the ω_2 region defined by the spectral FWHM (gray arrows). Interestingly, this form of the group delay describes the wavepacket shifting ~~shifts~~ to higher frequencies ~~from~~ ($300 \text{ cm}^{-1} \rightarrow 370 \text{ cm}^{-1}$ over $140 \text{ fs} \rightarrow 520 \text{ fs}$ in τ_d) in time. The blue-shift of the vibrational frequencies as a function of τ_2 is also revealed through a sliding window short time Fourier transform analysis (Supplementary Figure 4.2). We attribute the change in vibrational frequencies ~~describes~~ to the non-equilibrium relaxation of the excited wavepacket with respect to the ν_{Ru-N} coordinate measured experimentally during $0 < \tau_2 < 600$ fs. This dynamic blue-shift is suggestive of a rapid vibrational relaxation of the highly excited ν_{Ru-N} coordinate on $MLCT_B$ down the multidimensional anharmonic potential of the ν_{Ru-N} . Similar observations of blue-shifting of vibrational modes on photoexcited multidimensional surfaces have been made in organic complexes⁴⁴ and in reports of bridging cyanide ligand vibrational relaxation in mixed-valence complexes during photoexcited charge transfer.⁴⁵ Consistent with earlier studies,³⁴ the initial wavepacket’s spectral amplitude (Fig. 4, black) has largely disappeared by as ~~$\tau_2(\omega_2)$ approaches~~ 600 fs as shown by the absence

of an $\omega_2 = 340 \text{ cm}^{-1}$ peak in Fig. 2d and 2g which is consistent with earlier studies,³⁴ and it reflects the diminishing of the ρ_{35} density matrix element, triggering subsequent electronic delocalization.

- The following modifications have been made to the Supplementary Note 4 and the Supplementary Figures 4.1 and 4.2 to now read:

The initially excited state vibrational wavepacket composed of a coherence with the $\nu_{\text{RU-N}}$ mode is characterized by extracting the spectral phase of the ω_2 spectrum, $\phi(\omega_2)$, fitting the phase to an n^{th} order polynomial, and then obtaining the group delay, $\tau_d(\omega_2)$. Generally, a spectrum with arbitrary spectral phase variation can be written as $A(\omega - \omega_0) = |A(\omega - \omega_0)| e^{i\phi(\omega)}$. Here, the $|A(\omega - \omega_0)|$ factor is obtained by the absolute value of the Fourier transformed τ_2 -dependent peak intensities over $0 \leq \tau_2 \leq 600 \text{ fs}$ (i.e., $|FT(\tau_2)|$). The spectral phase and group delay are then obtained by⁷

$$\phi(\omega_2) = \tan^{-1} \left(\frac{\text{Im}[FT(\tau_2)]}{\text{Re}[FT(\tau_2)]} \right) \quad (1)$$

$$\tau_d(\omega_2) = - \frac{d\phi(\omega_2)}{d\omega_2} \quad (2)$$

The group delay describes the time-dependence of the frequency components in a wavepacket, which in this case reflects the propagation of the wavepacket during τ_2 in the excited MLCT manifold during the early time period, $0 \leq \tau_2 \leq 600 \text{ fs}$. We fit $\phi(\omega_2)$ over the spectral range of the ω_2 feature obtained in the FT analysis ($\omega_2 = 250\text{-}400 \text{ cm}^{-1}$) to polynomial functions of increasing order n , the coefficients ($P_{(n)}$) for each fit are given in Table SI 4.1. A meaningful description of the wavepacket propagation is obtained by observing the consistencies in $\tau_d(\omega_2)$, obtained by numerical differentiation of the polynomial function used in the fitting for different polynomial orders. The quartic polynomial ($n=4$) fit shown in Supplemental Figure 4e is used to obtain the group delay curve presented in manuscript Figure 4. The green curve in manuscript Figure 4 is identical to the red curve shown in Supplementary Figure 4f. As discussed in the manuscript, and shown in Supplementary Figure 4.1, all fits demonstrate consistent evidence for a blue shifting of the initially excited vibrational wavepacket during the first 600 fs of excited state relaxation. We place less emphasis on any interpretation based on the values of the coefficients for a particular fit given their dependence on the polynomial order, and instead consider the general trends observed across all fits. The same blue-shifting behavior is also observed through a short-time Fourier transform (STFT) analysis of the earlier time data, as shown in Supplementary Figure 4.2 below. We give preference to the group delay analysis shown in the manuscript over the STFT because the blue-shifting is occurring during the course of only a few cycles of the $\nu_{\text{RU-N}}$ vibration, which is also comparable to the $\sim 600 \text{ fs}$ lifetime of the wavepacket. Thus, an appropriate choice of windowing function becomes more difficult. Nevertheless, the consistency between the group delay analysis and the STFT confirms the blue-shifting of the $\nu_{\text{RU-N}}$ wavepacket during early times.

- The following figure has been added to the Supplementary Note 4:

Supplementary Figure 4.2 Short Time Fourier Transform Analysis of Initial Wavepacket. Analysis performed using the same windowing function used in the later time STFT analysis (double-sided tanh) discussed in more detail in supplementary note 6, but with a window FWHM of 500 fs that has τ_{center} delays varied through the corresponding earlier delay times $0 \leq \tau_2 \leq 600$ fs. The thick solid black line is a guide to the eye to highlight the blue-shifting of the $\nu_{\text{Ru-N}}$ center frequency. Measured in this way, the ω_2 vibrational frequencies blue-shift by $\sim 60 \text{ cm}^{-1}$ over the interval $250 \leq \tau_{\text{center}} \leq 570$ fs, consistent with the blue-shift measured by the spectral phase / group delay analysis in Supplementary Figure 4.1.

REVIEWERS' COMMENTS:

Reviewer #3 (Remarks to the Author):

I thank the authors for their careful consideration of our comments and now support the publication of the manuscript in its present form.